# The Effects of Soybean Meal on Growth, Bioactive Compounds, and Antioxidant Activity of *Hericium erinaceus*

**Preuk Chutimanukul** [1,*], **Siripong Sukdee** [1], **Onmanee Prajuabjinda** [2], **Ornprapa Thepsilvisut** [1], **Sumalee Panthong** [2], **Dusit Athinuwat** [1], **Wilawan Chuaboon** [1], **Phakpen Poomipan** [1] and **Vorapat Vachirayagorn** [1]

1   Department of Agricultural Technology, Faculty of Science and Technology, Thammasat University, Pathumthani 12120, Thailand
2   Department of Applied Thai Traditional Medicine, Faculty of Medicine, Thammasat University, Pathumthani 12120, Thailand
*   Correspondence: preuk59@tu.ac.th

**Abstract:** *Hericium erinaceus* (Bull.:Fr) Pers. is a medicinal mushroom that has various health benefits and is a rich source of bioactive compounds and antioxidant activity. In recent years, *H. erinaceus* has been considered for its many medicinal properties and is widely consumed in Asian countries. Remarkably, the effect of mushroom cultivation using substrates composed of soybean meal by-products on growth, as well as the enhancement of bioactive compounds and antioxidant activity, was evaluated. Our results confirmed that using soybean meal-produced *H. erinaceus* displayed a higher mycelial growth and biological efficiency than the control treatment. Bioactive compounds with triterpenoid content and total phenolic content of *H. erinaceus* grown on soybean meal contained the highest values at 56.78–69.15 mg Urs/g DW and 15.52–16.07 mg GAE/g DW, respectively, while *H. erinaceus* grown on the control treatment had the lowest value at 32.15 mg Urs/g DW and 7.75 mg GAE/g DW, respectively. In addition, *H. erinaceus* cultivated on soybean meal had higher DPPH activities than those grown on the control treatment, with $IC_{50}$ values of 0.67–0.89 and 1.08 mg/mL, respectively. Therefore, this study provided baseline information on the potential role of soybean meal by-product substrates in *H. erinaceus* growth and their effect on bioactive compounds and antioxidant activity.

**Keywords:** *Hericium erinaceus*; soybean meal; growth; bioactive compound; antioxidant activity





## 1. Introduction

Mushrooms are an increasingly common culinary choice since they are considered to contain beneficial nutrients and minerals [1,2]. Scientists have cataloged over 2000 different types of mushrooms, many of which are beneficial for medicine [3]. *Hericium erinaceus* (Bull.:Fr) Pers., also called Yamabushitake or lion's mane, belonging to the family Hericiaceae in the order Russulales [4], was the focus of this study since it is used in cooking as well as for medicinal purposes [5]. It is important to recognize that *H. erinaceus* is a source of bioactive substances which promote health, in addition to nutritious substances such as high-quality protein, fibers, vitamins, and minerals that can be ingested or taken orally [6,7]. Due to the fact that the fruitbody of *H. erinaceus* is composed of numerous components, it has been determined that it contains aromatic compounds, triterpene, sterols, hericenones, erinacine, and polysaccharides [7]. In addition, *H. erinaceus* has attracted the interest of numerous researchers in the past few years owing to its beneficial medical functions, including wound recovery properties, anticancer, antioxidant, antihypertensive, antidiabetic, and neurodegenerative disease potential, among other therapeutic potential [8,9].

In recognition of their significance in industry, agriculture, and health as a completely natural source of medicine, mushroom production has increased significantly over the past several decades and is expected to continue to do so in the future [2,10]. Currently, mushroom

cultivation imitates natural cultivation by utilizing materials where mushrooms can grow naturally, such as softwood sawdust or hardwood sawdust, as cultivation substrates. These substrates provide nourishment for the mushroom mycelium, allowing it to grow until it forms the fruiting body. Although substrates contain nutrients that mushrooms require, it is difficult to use them as the primary material in mushroom cultivation to produce high-quality products. Nitrogen is required for mushrooms to thrive; it is known that the source of the cultivation substrate [11], which has an effect on the growth of mycelia and the carbon-to-nitrogen (C/N) ratio, is a critical parameter affecting the biosynthesis of many metabolites in mushroom cultivation [12]. Furthermore, it is acknowledged that the substrate is the most critical part of mushroom development. In order to promote the growth of the mycelium, it is necessary to introduce additional nutrients, which will have an effect on the yield and quality of the mushrooms, as well as their ability to promote the synthesis of secondary metabolites with immunomodulatory and other biological properties.

Using plant by-products and wastes in the form of organic substances, such as soybean meal, provides a different approach to the cultivation of *H. erinaceus* to produce mushrooms. Soybean meal is a by-product of soy milk production, which has the advantage of a protein-rich substrate supplying nitrogen and minerals; it is a useful nutrient and is very important for mushroom cultivation in helping promote the growth of the mushrooms [13]. Consequently, the purpose of this study was to evaluate the application of soybean meal as a supplement in the cultivation of *H. erinaceus* to enhance its bioactive compounds and antioxidant activity.

## 2. Materials and Methods

### 2.1. Chemicals and Reagents

A concentration of 95% nitric acid and 99.8% glacial acetic acid was obtained from RCI Labscan Limited (Bangkok, Thailand); 60% perchloric acid was obtained from Sigma-Aldrich (St. Louis, MI, USA); 81.0–83.0% ammonium molybdate ($(NH_4)_6Mo_7O_{24} \cdot 4H_2O$)), 99% ammonium metavanadate ($NH_4VO_3$), 99% potassium dihydrogen phosphate ($KH_2PO_4$), 99% potassium chloride (KCl), 97.0% sodium hydroxide (NaOH), and 99.5% sodium carbonate ($Na_2CO_3$) were obtained from KemAus (Bangkok, Thailand); vanadate–molybdate reagent was obtained from Ricca chemical (Arlington, TX, USA). The standards and reagents of ursolic acid (3β-Hydroxyurs-12-en-28-oic acid), vanillin (4-Hydroxy-3-methoxybenzaldehyde), gallic acid (3,4,5-Trihydro-xybenzoic acid), Folin–Ciocalteu reagent, butylated hydroxytoluene (2,6-Di-tert-butyl-4-methylphenol), and DPPH (1,1-diphenyl-2-picrylhydrazyl) were purchased from Sigma-Aldrich (St. Louis, MI, USA). The organic solvents of 99.9% absolute ethanol ($C_2H_5OH$) and 99.8% absolute methanol (MeOH) were obtained from Merck KGaA (Darmstadt, Germany)

### 2.2. Analyses of the Physical Properties and Nutrient Composition of a Mushroom Cultivation Substrate

The substrate samples were prepared by oven-drying at 70 °C until a constant weight was achieved and sieved through a 2 mm sieve to analyze specific soil physical properties, such as the hydrogen ion concentration (pH), using a pH meter PC950, Apera Instrument, (Columbus, OH, USA), electrical conductivity (EC), using a conductivity meter Eutech CON 2700, Thermo Fisher Scientific (Waltham, MA, USA), organic carbon (OC), using a CHNS/O Analyzer model 628 series, Leco Corporation (St. Joseph, MI, USA), organic matter (OM) (organic carbon × 1.724), and C/N ratio (calculated by dividing the organic carbon value by the nitrogen value).

The nutrient composition was analyzed by applying the Udelhoven et al. [14] method, which consists of an analysis of total nitrogen (N), total phosphorus (P), and total potassium (K). The nitrogen in the materials was analyzed by using an elemental analyzer, the CHNS/O Analyzer; the phosphorus (P) was analyzed by using a UV-Spectrophotometer (UV-1280, Shimadzu, Japan) at the wavelength of 882 nm; and the potassium (K) was analyzed by using a flame photometer (410, Sherwood Scientific Ltd., Cambridge, UK).

### 2.3. Samples Preparation

Samples of fresh soybean meal obtained from soybean milk production were oven-dried at 70 °C until the soybean meal was completely dry, then blended in a commercial blender to be used as a substrate in the experiment.

The fruiting body of *H. erinaceus* was cultivated in Pathumthani (Thailand) in September 2022 on a substrate containing rubberwood sawdust (RS) and soybean meal (SM) in mass ratios of 80:0 (control), 78:2, 76:4, 74:6, 72:8, and 70:10. The same amounts of corn stalk, fine bran, and lime were added to each treatment (Table S1), and 750 g of each were contained in plastic bottles (a total of 5 replicates were cultured, 2 bottles per replicate). After that, they were autoclaved at 120 °C for 30 min at 15 pounds per square inch and allowed to settle at room temperature for 24 h. The *H. erinaceus* inoculum was then added to the substrate-containing bottle. The inoculated plastic bottles were placed in a greenhouse with sufficient airflow at a temperature of 20 ± 2 °C and 80 ± 5% relative humidity until the mycelium was fully developed for each treatment; then, the lids were opened and left for 18 days, when the first generation of mushrooms was harvested.

At the end of the harvesting phase, mushrooms were collected for all the treatments, and the mycelial growth (days), the diameter of the cap (cm), and the biological efficiency (%), based on Equation (1) below, were calculated:

$$\text{Biological efficiency (\%)} = \frac{\text{Fresh weight of fruiting bodies (g)}}{\text{Dry weight of substrate after harvest (g)}} \times 100 \qquad (1)$$

### 2.4. Preparation of Extracts

The fresh fruitbodies were oven-dried at 65 ± 2 °C for 72 h, then blended in a commercial blender and stored in sealed containers before analysis. After that, the mushroom powder (10 g) from the oven-dried fruitbodies of *H. erinaceus* was soaked in 50 mL of 95% ethanol at room temperature. The extracts were filtered through Whatman No. 1 filter paper (the extraction was repeated every 3 days for 9 days). A rotary evaporator was used to decant, filtrate, and then concentrate the solvent containing the distillate under a vacuum condition (R-300, BUCHI, Switzerland). Consequently, the extracts were calculated as the % yield of *H. erinaceus*, which was based on Equation (2) below:

$$\text{Yield (\%)} = \frac{\text{Weight of dry extract (g)}}{\text{Weight of dry plant material (g)}} \times 100 \qquad (2)$$

### 2.5. Determination of Total Triterpenoid Content

The total triterpenoid content in the *H. erinaceus* ethanol extracts was determined according to the method of Ni et al. [15], with some modifications, and using ursolic acid as a standard. A calibration curve was generated using a solution at concentrations of 1000, 500, 250, 120, 62.5, 31.25, and 16.625 µg/mL in methanol. In short, 10 mg of each *H. erinaceus* extract was dissolved in 1 mL of methanol, followed by the transfer of 300 µL of each extract to a test tube. Afterward, 50 µL of vanillin–acetic acid solution (5:95, *w/v*) and 800 µL of perchloric acid were added, and the mixture was incubated for 15 min at 60 °C in a water bath. The mixture solution was then moved into an ice water bath. Finally, 5 mL of acetic acid was added, and the mixture was set aside for 15 min at an ambient temperature. The absorbance at 548 nm was measured with a microplate reader (Multiskan GO, Thermo Scientific, Waltham, MA, USA), and the blank solution was used as a standard. The process was conducted in triplicate, and the results were presented as milligram equivalents of ursolic acid (Urs) per gram of dry weight (mg Urs/g DW).

### 2.6. Determination of Total Phenolic Content

The content of total phenolic compounds in the *H. erinaceus* ethanol extracts was determined by the Folin–Ciocalteu method [16], as applied by Miliauskas et al. [17], for the preparation of the calibration curve; 20 µL of ethanol–gallic acid solutions at concentrations

of 200, 160, 80, 40, 20, 10, and 5 μg/mL were mixed with 80 μL of 7.5% sodium carbonate and added to 100 μL of Folin–Ciocalteu reagent (dilution 1:10) and 80 μL of 7.5% sodium carbonate, in the last step. After 30 min, the absorption was measured at 765 nm by using a microplate reader, and the calibration curve was drawn. Each extract was mixed with 20 μL of absolute ethanol and the same reagents as described above. The total phenol content was stated as the number of milligram equivalents of gallic acid (GAE) per gram of dry weight (mg GAE/g DW).

### 2.7. Scavenging Activity on DPPH Radicals

The radical scavenging activity of the *H. erinaceus* ethanol extracts was measured using the method, with some modifications, from Seephonkai et al. [18] The samples were prepared at the following concentrations: 3, 2, 1, 0.5, and 0.25 mg/mL in absolute ethanol. A 100 μL of the sample was mixed with 100 μL of DPPH radicals. After 30 min at room temperature in a dark place, the absorbance was measured at 520 nm by using a microplate reader, and the percentage of inhibition was calculated using the following Equation (3):

$$\text{Inhibition (\%)} = [(A_{DPPH} - A_S)/A_{DPPH}] \times 100 \tag{3}$$

where $A_{DPPH}$ is the absorbance of the DPPH solution, and $A_S$ is the absorbance of the solution containing the sample.

The half maximal inhibitory concentration ($IC_{50}$) value indicates the antioxidation activity. Butylated hydroxytoluene (BHT) was used as the positive control, while the absence of mushroom extract was used as the negative control. The results were presented as milligrams to milliliters.

### 2.8. Statistical Analysis

The results were expressed as the mean $\pm$ standard deviation (SD). The experimental data were analyzed using a one-way analysis of variance with Duncan's multiple range test. The statistical significance of the mean differences was based on a *p*-value of <0.05 using IBM SPSS Statistics 21. The data sets presented here are chosen from several trials.

## 3. Results and Discussion

By-products of soybean meal show highly significant variations between individual samples and are therefore difficult to standardize. It has been reported that soybean meal contains 7.77 to 8.00% nitrogenous compounds [19], which is an important factor affecting the growth and the amount of antioxidants in mushrooms.

### 3.1. Study of the Physical Properties and Nutrient Composition of the Mushroom Cultivation Substrate

An analysis of the physical properties of the *H. erinaceus*-growing substrate revealed, as shown in Table 1, that the pH level ranged from 5.93 to 7.77. The pH of the medium is a crucial intrinsic factor, as it influences the ionic state of the medium, the structure, morphology, and physiological functions of the fungal cells, as well as the nutrient absorption and product biosynthesis [20]. Moreover, this will affect mushroom growth and fruiting. The mushroom will only develop hyphae and produce a fruiting body due to the acidic substrate. According to research by Imtiaj et al. [21], *H. erinaceus* grows effectively at pH levels between 5 and 9, with the optimal value at pH 6. The electrical conductivity (EC) of soybean meal, which was added to all the treatments, ranged from 1.31 to 1.72. dS m$^{-1}$. When the EC value of the control treatments was 2.07 dS m$^{-1}$, the low EC resulted in excellent agronomic behavior. It was discovered that there is a threshold in EC at 1.6 dS m$^{-1}$ over which a significant decrease in yield may be observed [22]. For the substrate to be suitable for facilitating the growth and production of mushroom mycelia, it must be amended to a low EC [23]. In addition, the organic carbon content and organic matter content of the mushroom-growing substrate were determined to be between 46.0 and 46.73% and 79.31 and 80.56%, respectively. Carbon sources extracted from rubberwood sawdust are mostly cellulose and hemicellulose, which are carbohydrates with large and difficult-to-decompose molecules [24]. Thus, microorganisms

are necessary for their breakdown into smaller molecules. The mycelium then extracts the organic matter and organic carbon content in order to generate new microbial cells during the fermentation process, thereby releasing the nutrients necessary for mushroom growth.

**Table 1.** Physical properties and nutrient composition of *H. erinaceus* substrate before cultivation.

| Treatments (*w/w*) | Physical Properties | | | | Nutrient Composition | | | C/N Ratio |
|---|---|---|---|---|---|---|---|---|
| | pH | EC (dS m$^{-1}$) | Organic Carbon (%) | Organic Matter (%) | Total N (%) | Total P (%) | Total K (%) | |
| RS80:SM0 (control) | 5.93 | 2.07 | 46.73 | 80.56 | 0.65 | 0.10 | 0.51 | 71.78 |
| RS78:SM2 | 7.62 | 1.72 | 46.47 | 80.11 | 0.74 | 0.09 | 0.58 | 62.81 |
| RS76:SM4 | 7.77 | 1.39 | 46.00 | 79.30 | 0.76 | 0.10 | 0.61 | 60.53 |
| RS74:SM6 | 7.55 | 1.74 | 46.59 | 80.32 | 0.70 | 0.10 | 0.53 | 66.51 |
| RS72:SM8 | 7.76 | 1.31 | 46.28 | 79.78 | 0.77 | 0.10 | 0.63 | 59.74 |
| RS70:SM10 | 7.44 | 1.49 | 46.35 | 79.91 | 1.25 | 0.10 | 0.61 | 37.14 |

RS = Rubberwood Sawdust and SM = Soybean Meal. This table is not a statistical analysis.

According to an analysis of the nutrient composition, as shown in Table 1, it was found that the total nitrogen content of the soybean meal added to the treatments ranged from 0.70 to 1.25%. This amount was suitable for the growth of mycelium and provides an essential source of energy for the formation of mushroom cell structures. While the total nitrogen content of the control treatments was 0.65%, the addition of a nitrogen source supplement to the substrate will increase the nitrogen content to a sufficient level for mycelium growth. In addition, nitrogen in the substrate may also interact with carbon at the ratio of C/N. Critical processes, such as lignocellulosic degradation, depend on the C/N ratio of the substrate [25] and are very important in consideration of their role in mushroom growth [23]. It was determined that the C/N ratio of the treatments that included soybean meal ranged between 37.14 and 66.51. While the control treatment had a C/N ratio of 71.78, the high value was caused by the substrate's high carbon and low nitrogen content. Therefore, the addition of soybean meal will alter the C/N ratio to a range that is more beneficial to mushroom cultivation. In addition, the research has shown that *Flammulina filiformis* can degrade substrates with a low ratio of C/N much more effectively than those with a high ratio of C/N [26]. Additionally, the total phosphorus and total potassium contents of the mushroom culture substrate were determined to be between 0.09 and 0.10% and 0.51 and 0.63%, respectively. Phosphorus and potassium are essential nutrients for mushroom growth, although in small amounts; they still influence the growth of *H. erinaceus* hyphae since they promote the normal development of mycelium's physiological processes.

### 3.2. Growth of H. erinaceus

The mycelial growth of the *H. erinaceus* cultivars on various substrates on the days tested varied significantly ($p < 0.05$). The results are shown in Table 2. It was found that the treatment with soybean meal resulted in a faster growth rate of *H. erinaceus* mycelium than the control treatment, based on an average mycelial growth time of 15.70 to 18.00 days, while the control treatment required an average of 19.30 days. The study also revealed that the addition of soybean meal increased the growth rate of *H. erinaceus* mycelia by triggering the release of nitrogen from degraded organic matter. In addition, a report by Harith et al. [26] found that a substrate with a low nitrogen content, compared to a substrate with a high nitrogen content, could boost the mycelium growth rate of *Flammulina* spp. The improved proliferation of mycelia suggests that the substrates contain an appropriate C/N ratio [27] caused by solid-state cultivation, which directly influences the growth of mushrooms [28,29]. Furthermore, the diameters of the mushrooms were measured (Figure 1A–F), and statistically significant differences ($p < 0.05$) were found. The addition of soybean meal to the treatment increased the diameter to a maximum of 9.36 to 10.78 cm, while the control treatment had the smallest average diameter, which was 8.2 cm.

An increase in mushroom diameter may be attributable to the addition of soybean meal to the substrate, as mushrooms require nitrogen in the form of organic matter, which is taken in from the substrate as a protein source and increases nitrogen levels. This also resulted in a C/N ratio within the optimal range for the growth of *H. erinaceus*. According to the report by Ashrafi et al. [30], the C/N ratio plays an integral part in the growth of, and is essential for, mushroom cultivation, affecting mushroom yield and development. Thus, an accurate analysis of carbon and nitrogen content in a substrate and effective utilization of a C/N ratio would increase the mycelial growth rate and cap diameter [31].

**Table 2.** Growth parameters of the *H. erinaceus* cultivation on various substrates.

| Treatments (*w/w*) | Mycelial Growth (Day) | Diameter of Cap (cm) | Biological Efficiency (%) | % Yield |
|---|---|---|---|---|
| RS80:SM0 | 19.30 ± 0.45 [d] | 8.32 ± 0.47 [c] | 15.57 ± 0.71 [d] | 14.94 ± 0.30 |
| RS78:SM2 | 18.00 ± 0.35 [c] | 9.36 ± 0.42 [b] | 27.07 ± 1.03 [c] | 12.88 ± 1.34 |
| RS76:SM4 | 16.30 ± 0.45 [ab] | 10.56 ± 0.40 [a] | 35.60 ± 1.08 [a] | 14.80 ± 1.73 |
| RS74:SM6 | 17.80 ± 0.57 [c] | 9.58 ± 0.34 [b] | 27.72 ± 1.00 [c] | 13.58 ± 0.83 |
| RS72:SM8 | 16.90 ± 0.22 [b] | 10.36 ± 0.32 [a] | 32.86 ± 1.26 [b] | 12.74 ± 0.29 |
| RS70:SM10 | 15.70 ± 0.84 [a] | 10.78 ± 0.28 [a] | 33.27 ± 1.13 [b] | 14.24 ± 2.19 |
| F-test | ** | ** | ** | ns |
| C.V.% | 2.81 | 3.38 | 2.74 | 7.98 |

Values are means with standard deviations (*n* = 5). Means with different letters in the same column are significantly different by Duncan's multiple range tests (*p* < 0.05). ** There were significant differences at *p* < 0.01, and ns = not significant.

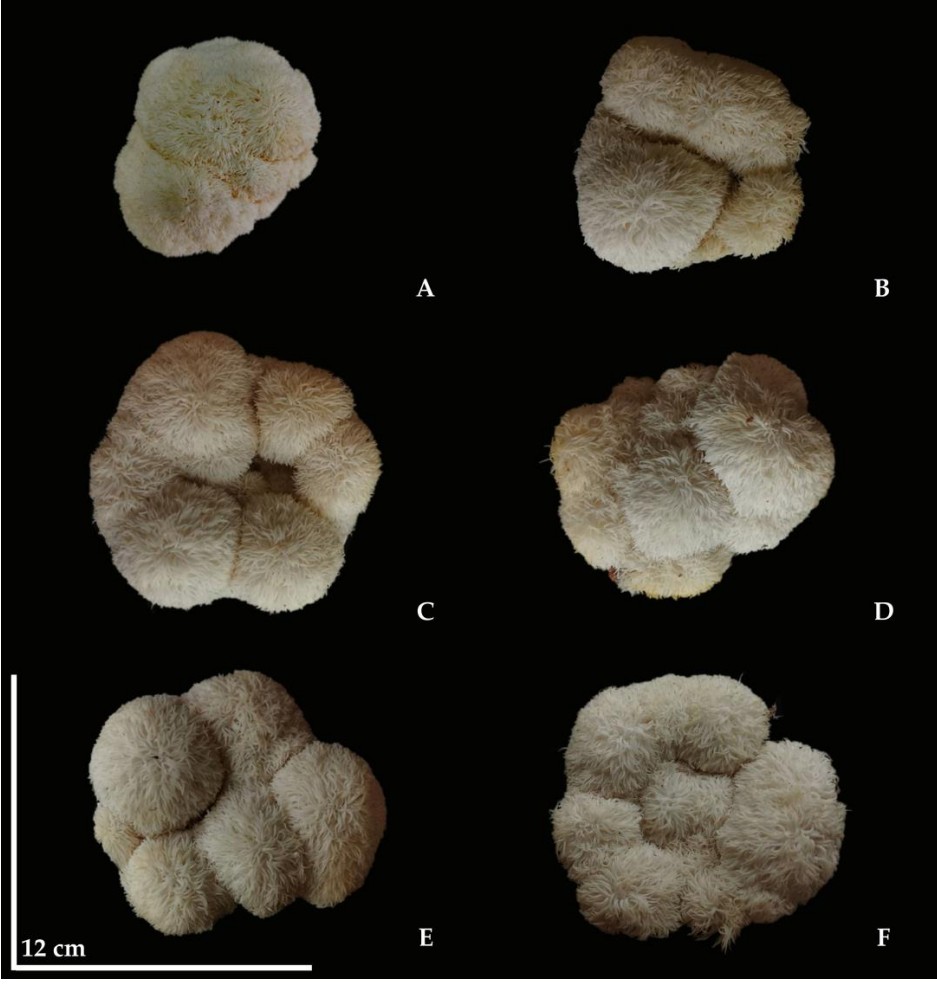

**Figure 1.** Diameter of *H. erinaceus* cultivated on various substrates: (**A**) RS80:SM0 (control), (**B**) RS78:SM2, (**C**) RS76:SM4, (**D**) RS74:SM6, (**E**) RS72:SM8, and (**F**) RS70:SM10.

The biological efficiency of *H. erinaceus* is shown in Table 2. The result shows that the addition of soybean meal resulted in a biological efficiency between 27.07 and 35.60%, whereas the control treatment had a biological efficiency of 15.57%, which indicates a statistically significant increase in biological efficiency compared to the control treatment ($p < 0.05$). The use of soybean meal as a supplement, which supports the growth of mycelium and mushroom fungi, will increase production efficiency, according to the study. In accordance with the findings of Wang et al. [32], it is stated that nutrient supplements are essential for mushroom cultivation, as they increase the quantity and quality of the mushrooms, particularly those with levels of nitrogen. Since nitrogen is an essential component of protein, it helps promote the growth of mushroom mycelium and increases the efficiency of mushroom production. In line with research conducted by Rodrguez Estrada et al. [33], by supplementing the substrate with a commercially available delayed-release nutrient (corn and soybean-based), the biological efficiency (BE) of Pleurotus eryngii var. eryngii was enhanced. In addition, according to Royse's [34] study, the addition of soybean meal to P. sajor-caju's planting mixture contributed to a 79% increase in biological efficacy compared to the absence of soybean meal. For the % yield analysis shown in Table 2, it was found that the % yield of *H. erinaceus* extraction using the conventional maceration method was 12.74 to 14.94% using 95% ethanol for extraction; there was no difference in the extracted yield. In general, the preliminary solvent extraction yields a crude extract, which is the mushroom extract; if the same solvent is used for fermentation, where the solvent is exposed to the mushroom powder of *H. erinaceus* until the solvent was inserted and extracted, the obtained the yield of *H. erinaceus* was no different.

### 3.3. Determination of Total Triterpenoid Content of H. erinaceus

The analysis of total triterpenoid content found in ethanolic extracts from *H. erinaceus* is presented in Figure 2. This shows that the total triterpenoid content of mushroom extracts grown on the substrates of soybean meal was significantly greater than that of mushroom extracts grown on the treatment control substrates ($p < 0.05$). The triterpenoids in the treatments with added soybean meal ranged from 56.78 to 69.15 mg Urs/g DW, whereas the triterpenoids in the control treatments were 32.15 mg Urs/g DW. It was shown that the use of soybean meal substrate in all the treatments may increase the triterpenoid content of *H. erinaceus* by improving the nutrient accessibility of the soybean meal [35]. To our knowledge, this is the first report to describe the ability to enhance the content of triterpenoids in *H. erinaceus* by adding soybean meal. Our results confirmed that the synthesis of triterpenoids from *H. erinaceus* extracts is stimulated by nitrogen in the form of organic nitrogen in the substrate, supporting mushroom growth and the biosynthesis of secondary metabolites [36–39]. Most of the triterpenoids from mushrooms have important medicinal properties. An important step toward enhancing the production of triterpenoids in mushrooms is to understand the mechanism of triterpenoid biosynthesis in this mushroom. Previous studies found that the triterpenoid content is synthesized via the mevalonic acid (MVA) pathway, which exhibited a positive correlation between gene expression and changes in the accumulation of triterpenoids in mushrooms during development [40]. The nitrogen contained in soybean meal may affect the expression of genes involved in the triterpenoid biosynthesis pathway, which is affected by the nitrogen form and amount in substrates [39], in which the decomposition of available organic nitrogen for energy to promote growth and development results in an increased content of triterpenoids by enhancing its absorption into microbial biomass as the primary mechanism by which nutrients are released to the mycelium [41]. Moreover, some hypotheses have also been presented to explain the potential trade-off between the synthesis of secondary metabolites, according to previous research conducted by Darmasiwi et al. [42], in which mycological assays of *H. erinaceus*-cultivated products obtained from the commercial basidiome contained small amounts of terpenoids. In contrast, Yang et al. [35] revealed bioactive compounds by adding ground soybean meal to *Ganoderma lucidum* substrates. It was discovered that the triterpenoid content of *G. lucidum* extract was greater when compared to the control. Accordingly, the total triterpenoid content of medicinal and edible mushrooms varies

based on the substrate used, which suggests that the variability of total triterpenoid content in cultivated mushrooms is possibly dependent on the substrate composition. Also, the substrate is the most important intrinsic factor, as it is the sole source of essential nutrients required for mushroom growth and development, from which the mushrooms catabolize complex organic compounds in the soybean meal substrate, assimilate minor nutrients through their mycelia, and stimulate the biosynthesis of all the triterpenoids in *H. erinaceus* [27].

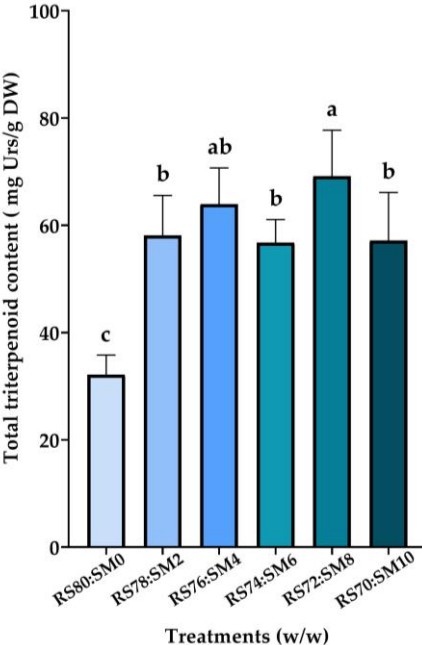

**Figure 2.** Total triterpenoid content of *H. erinaceus* cultivated on various substrates. Values are means with standard deviations (*n* = 5). The different letters above the bars indicate statistically significant differences by Duncan's multiple range tests (*p* < 0.05).

There is a vast variety of biosynthetic pathways involving triterpenoids, and they are found in many plant species. It is the major compound of some traditional medicinal herbs and is well-known to possess a wide range of biological functions. Triterpenoids are synthesized in a great number of organisms and are especially abundant and diverse in plants, including those that can be found in *H. erinaceus* [43]. Triterpenoids have long been considered to have a number of biological effects [44]. Presently, the more-than-exponential increase in the number of reports regarding bioactive triterpenoids in the last decade reflects their growing importance as sources of medications and preventive medicines. In fact, some are increasingly being used for medicinal purposes for a variety of clinical diseases in many Asian countries. Of these, triterpenoids are compounds that have attracted considerable interest, based on their remarkable antioxidative, anti-inflammatory, and anticancer biological functions [44,45]. Moreover, several triterpenoids have been reported to have inhibitory activity against the human immunodeficiency virus [40].

*3.4. Determination of Total Phenolic Content of H. erinaceus*

The analysis of total phenolic content found in the ethanolic extracts from the *H. erinaceus*, shown in Table 3, indicates that the ethanolic extracts of *H. erinaceus* from the soybean meal-added treatment contained phenolic compounds in the range of 15.26 to 16.07 mg GAE/g DW (or a two-fold increase), whereas the control treatment had the lowest concentration of phenolic compounds at 7.75 mg GAE/g DW. Our results confirmed that the addition of soybean meal resulted in no difference in total phenolic content, but the results indicated a statistically significant increase in phenolic compounds compared to the control treatment (*p* < 0.05). Ordinarily, variations in mushroom composition are linked to the use of various cultivation substrates, e.g., the addition of soybean meal leads to an increase in secondary

metabolites, which could be attributed to its higher nitrogen content [46–48]. Similarly, Shetty [49] found that substrate composition could potentially stimulate the biosynthesis of more phenolic compounds in mushrooms, resulting in variations in total phenolic content based on the chemical composition of the growth substrates, for the phenolic compounds increased along the growth time. This is probably a response to the nitrogen in soybean meal during the growth of the mycelium by the absorption of nutrients used in the synthesis of phenolic compounds. Generally, the organic sources of nitrogen generate the synthesis of a higher quantity of phenolic compounds [50]. Therefore, the higher total phenolic content of *H. erinaceus* extract grown with soybean meal might account for the better results found in their control treatment, which had a low nitrogen content. This outcome corresponds with previous findings by Singh et al. [51] that analyzed the total phenolic content of oyster mushrooms by adding desiccated soybean plants to the culture substrate in order to stimulate phenolic synthesis. According to Yang et al., [35] the total polyphenol content of *Lentinula edodes* fermented on okara, a by-product of soybean processing, was evaluated. After fermentation, it was discovered that the total polyphenols content of *L. edodes* was 7 mg GAE/g DW, a nine-fold increase over the control levels of 0.8 mg GAE/g DW. Analysis of the total phenolic content in *H. erinaceus* derived from commercial basidiome by Darmasiwi et al. [42] revealed that the total phenolic content of *H. erinaceus* extracts was 9.96 mg GAE/g DW. The findings align with those found in other studies on the same topic; for instance, Marimuthu et al. [7] found that *H. erinaceus* contained 8.77 mg GAE/g of total phenolic content.

**Table 3.** Total phenolic content and antioxidant activity of *H. erinaceus* cultivated on various substrates.

| Treatments (*w/w*) | Total Phenolic Content (mg GAE/g DW) | Antioxidant Activity DPPH Scavenging Activity (IC$_{50}$, mg/mL) |
|---|---|---|
| RS80:SM0 | $7.75 \pm 5.06$ [b] | $1.08 \pm 0.23$ [c] |
| RS78:SM2 | $15.26 \pm 9.75$ [a] | $0.84 \pm 0.13$ [b] |
| RS76:SM4 | $16.07 \pm 3.54$ [a] | $0.67 \pm 0.04$ [a] |
| RS74:SM6 | $15.52 \pm 8.40$ [a] | $0.89 \pm 0.06$ [b] |
| RS72:SM8 | $15.62 \pm 9.54$ [a] | $0.80 \pm 0.10$ [ab] |
| RS70:SM10 | $15.59 \pm 4.90$ [a] | $0.73 \pm 0.04$ [ab] |
| F-test | ** | ** |
| C.V.% | 4.96 | 10.91 |

Values are means with standard deviations (*n* = 5). Means with different letters in the same column are significantly different by Duncan's multiple range tests (*p* < 0.05). ** There are significant differences at 99%.

The interest in extracts containing bioactive compounds has steadily increased in recent years due to the demand for products with a direct effect on human health. It is desirable to control the negative effects of free radicals. Among the existing natural antioxidants in mushroom extracts, the phenolic compounds are known to exert a direct inhibitory effect on free radicals [50,52]. The effect is manifested through their chemical structure, in particular by the number of hydroxyl groups, which is directly proportional to antioxidant activity, and by their location in the molecule. These compounds reduce the risk of major chronic diseases and have antitumor effects [50,53].

### 3.5. DPPH Radical Scavenging Activity

The antioxidant properties of ethanolic extracts from *H. erinaceus* assayed by the DPPH radical scavenging method were compared with standardized BHT (IC$_{50}$ value of 12.03 µg/mL). In general, the antioxidant properties are inversely proportional to their IC$_{50}$ values, with lower values indicating greater antioxidant activity. The results are shown in Table 3, indicating that the mushroom growth substrate affected the DPPH radical scavenging activities. The extracts of *H. erinaceus* grown on soybean meal with the IC$_{50}$ value between 0.67 and 0.89 mg/mL were significantly more effective at scavenging DPPH radicals than the control treatment with the IC$_{50}$ value of 1.08 mg/mL (*p* < 0.05). Yan et al. [54] reported various solvent extractions of *H. erinaceus* with IC$_{50}$ values ranging from 1.07 to

1.62 mg/mL for DPPH radical scavenging activity, indicating that the hydrogen-donating capacities of *H. erinaceus* extracts grown on soybean meal substrates were elevated. The wide variety of bioactive compounds detected in *H. erinaceus* extracts is typically associated with their functional properties. Due to their redox properties, phenolic compounds can function in the human body as reducing agents, hydrogen donors, and singlet oxygen quenchers, thus serving as excellent candidates for antioxidant activities [55,56].

Therefore, the total phenolic content and antioxidant activity of *H. erinaceus* grown on soybean meal substrates were greater than those grown on sawdust-only substrates. The nitrogen in soybean meal results in a higher total phenolic content, which might account for the results found in their antioxidant activity. Similarly, Yildiz et al. [56] reported that the mushroom's total phenolic content and antioxidant activity were related to the substrate on which it was grown. This study demonstrated that the substrates of soybean meal affected the bioactive compounds and antioxidant properties of *H. erinaceus*, due to mushrooms' ability to assimilate nutrients from soybean meal substrates for their development and growth. It is well established that the development phase of mushrooms influences their bioactive substances and a variety of vital biological processes [57]. However, the addition of soybean meal resulted in an increase in the amount of bioactive compounds and antioxidant activity that were not significantly different, because the physical properties and nutrient composition of the substrate involved in the growth and synthesis of the bioactive compound and antioxidant activity of *H. erinaceus* were very similar.

## 4. Conclusions

This study demonstrated that by-products of soybean meal can serve as viable substrates for the cultivation of *H. erinaceus*, and that substrate RS76:SM4 has the potential to be used for the production of *H. erinaceus* because it has the highest biological efficiency. Similarly, it has been proven that mushroom cultivation using soybean meal could represent a novel and cost-effective method for producing mushrooms abundant in growth-promoting nutrients. Moreover, compared to the control treatment, mushrooms with substantially increased antioxidative potential have been found to contain significantly greater amounts of bioactive compounds in relation to the total triterpenoid and phenolic contents, as the physicochemical properties of the cultivation substrate are in the appropriate range. In addition, it was observed from the nutrient composition analysis that the higher nitrogen and potassium concentration in the substrate was especially beneficial for the growth and development of *H. erinaceus*. The results obtained in this study suggest that *H. erinaceus* acquires beneficial nutrients from soybean meal substrate as a growth supplement and suggest its use in the production of bioactive compounds in medicinal plants.

**Supplementary Materials:** The following supporting information can be downloaded at: https://www.mdpi.com/article/10.3390/horticulturae9060693/s1, Table S1: The percentage ratio of cultivation mixture by weight of *H. erinaceus* spawns in the experiment.

**Author Contributions:** Conceptualization, P.C.; methodology, P.C., and O.P.; validation, P.C., S.S., and O.P.; formal analysis, P.C., O.P., O.T., and S.P.; investigation, P.C., and S.S.; resources, P.C., O.P., and S.S.; data curation, P.C.; writing—original draft preparation, P.C., O.P., and O.T.; writing—review and editing, P.C., O.P., and S.P.; supervision, P.C., and D.A.; project administration, P.C., W.C., P.P., and V.V.; funding acquisition, P.C. All authors have read and agreed to the published version of the manuscript.

**Funding:** This research was funded by Thammasat University Research Fund, Contract No. TUFF 06/2565.

**Data Availability Statement:** The original contributions and data presented in this research are included in the article; further inquiries can be directed to the corresponding authors.

**Acknowledgments:** The authors would like to thank Thammasat University Center of Excellence in Agriculture Innovation Centre through Supply Chain and Value Chain, Faculty of Science and Technology, Faculty of Medicine, Thammasat University, for providing technical support and instruments. We are also grateful to Fresh & Friendly Farm Co., Ltd., Thailand for providing the mushroom

production facility and instruments. We would like to thank Green Spot Co., Ltd., Thailand for the materials.

**Conflicts of Interest:** The authors declare no conflict of interest. The funders had no role in the research, in the analyses or the interpretation of data, in the writing of the manuscript, or in the decision to publish the results. This research was conducted in the absence of any commercial relationships that could be construed as a potential conflict of interest.

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
