# Peer review of "The Effects of Soybean Meal on Growth, Bioactive Compounds, and Antioxidant Activity of Hericium erinaceus"

_horticulturae, doi:10.3390/horticulturae9060693_

Round 1

Reviewer 1 Report

The subject of the manuscript “The Effects of Soybean Meal on Growth, Bioactive Compound 2 and Antioxidant Activity of Hericium erinaceus” is relevant but requires critical revision and appropriate rewriting before further consideration for publication.  Corrections must also be made in the abstract.

Some comments:

I miss a more accurately and detailed description of methods and results. Authors are mainly focused on describing references data.

The Introduction lacks logical consistency. For example, do you think mushrooms are blooming? Please correct this blunder. (48 L)

What are "they"?  "..it is difficult to use them as the primary  material " please specify (49 L).

Methods

Please reconsider the sentence in 50-53 L.

The information from 96 L should be in the Introduction.

Please change the title of 2.3.

How many repetitions of each treatment were prepared? How many days the experiment last?

The description of used growing substrate treatments and control must be accurately presented in methods and not repeat in the results section.  

Please clarify what means “the extracts were calculated as the %yield of H. erinaceus” and how it was made,

Please clarify: “with some modifications of ursolic acid as a Standard“ (120 L).

What equipotent was used for absorbency measurements?

Please indicate in which units was expressed antioxidation activity.

What abbreviations BHT means?

The information on used data sets must be presented in 2.8. (“The data sets present here are chosen from several trials“ ).

 Results

Results description is complicated and inconsistent. Please describe your findings sequentially, and then compare and discuss with literature findings.

The introduction to description of results is no clear. Who was reported? (161-163 L)

No such explanation is needed for pH level like „was moderately acidic to slightly alkaline“ (168-169)

The title of Table 1 must be reconsidered. What do you mean „after cultivation“?

In titles of 3.2. and 3.3. „Determination“is illogical and should be changed.

„The %Yield of H. erinaceus extracts ranged from 12.74 to 14.94. These outcomes may be due to the use of 95% ethanol in all H. erinaceus extracts“. What do you mean, please clarify. In methods there was no description of this parameter determination. (254-257 L)

Interpretation of the results on triterpenoid content is lacking. For example, what could account for the differences in triterpenoid content using different substrates (Fig. 2).

Pease reconsider: „The synthesis of triterpenoids from H. erinaceus extracts requires nitrogen in the form of organic matter in the substrate for mushroom growth and biosynthesis“as it is it is scientifically incorrect.

The description of the used method (301-304 L) is assigned to the methods section.

What do you mean: “Total phenolic content in H. erinaceus from growth on various substrates” (305).

Please clarify the statement: “The increase in total phenolic content appears to be related to the selective absorption of nutrients by cultivation substrates” (328-329) as it contradicts the results obtained. In your experiment, the amount of phenolic compounds did not depend on the differences in the substrates used, except rom the control.

In Table 3 do not duplicate: “Antioxidant activity DPPH scavenging activity”.

The conclusions must be reconsidered as the effect of different substrate treatments on the accumulation of compounds has not been revealed.  

It was not evaluated micro elements.

Please use the full Latin name with species authors when you first mention it in the Abstract and Introduction

 Don't start a sentence like that „95% nitric acid“ (67 L).

Please use lowercase letters „Electrical Conductivity“, „Hydrogen ion Concentration“.

Pleurotus eryngii var. Eryngii, P. sajor-caju etc.

Please note that the presented comments do not include all inaccuracies, but are for illustrative purposes only.

Weak English leads to confusion and limitations in the description of methods and results.  

Reviewer 2 Report

The manuscript entitled ”The Effects of Soybean Meal on Growth, Bioactive Compound and Antioxidant Activity of Hericium erinaceus" is interesting and important for improving mushroom production by improving the medium using soybean meal. However, there are some points necessary to be revised.

Please add an explanation of the characteristics of the soybean meal used for this experiment. Is this soybean meal imported? Or is it the residue of oil extraction? 

What is the advantage to use soybean meal compared with other waste organic matter? 

The authors suggested that the improvement by the addition of soybean meal decreased the C/N ratio of the medium. However, for this purpose, you can add some chemical N such as urea, or ammonium. Is this beneficial?

The authors show the total content of triterpenoid and phenolic compounds. Please show the example of triterpenoids and phenolic compounds in this mushroom based on the literature. In addition, please explain the advantage and disadvantages of these compounds for health. Some of them may be toxic, and some are beneficial.

Materials and methods: Please add the section explaining the origin and characteristics of soybean meal first.

Please add the Table S1 in the text. 

Line 129: Please add equivalent for "milligram equivalents of ursolic acid (Urs) per gram of dry 129 weight (mg Urs/g DW)"

Line 151: Please add full spells for BHT.

Table 1 total N content is not depended on the percentage of soybean meals. The total N contents of RS74 and 76 were lower than that of RS 72. Please confirm the table.

Please use italic for the scientific names, such as H. erinaceus.

As a conclusion, which proportion of soybean meal do the author recommend for the cultivation of Hericium erinaceus?

Relatively good.

Reviewer 3 Report

Lecture of ‚The Effects of Soybean Meal on Growth, Bioactive Compound and Antioxidant Activity of Hericium erinaceus’ was an interesting read. Manuscript describes influence of soybean meal addition on some parameters of Hericum erinaceus.

 This is a list of my remarks:

 Line 12-13: ‚In recent years, the cultivation of H. erinaceus has been considered as alternative food supplement’ I’m not sure if we can talk that cultivation of H. erinaceus may be considered as alternative supplement. What about wild mushrooms of this species?

 Please put values of IC50 from L22 abstract into one (second) pair of brackets, placed at the end.

 L 46-47 ‚plant waste materials, such as wheat,’ please clarify/justify using word ‚wheat’

 part 2.4 why extraction was conducted for such long time? Drying was done in 65oC, so it seems that extraction could also be executed faster

 part 3.0 please report table number at the beginning of each paragraph

 In part 2.8 it is stated: ‚The results were expressed as mean ± standard deviation (SD). All experimental data were analyzed using one-way analysis of variance with Duncan’s multiple range test.’

however Table 1 lacks S.D. values and designation of statistical groups.

 L321 lowercase for okara

 part 4 Conclusions must be claryfied. e.g. ‚a novel and cost-effective method for producing mushrooms abundant in growth-promoting nutrients’. can you specify what growth-promoting nutrients in mushroom extracts were evaluated during this study? Please discuss in part 4 or 3 the fact that even small addition of soymeal modulates the total phenolics content and DPPH IC50 value and it is not dependent of the percentage of soymeal addition (vide > table 3).

 L369 - ‚beneficial nutrients’ all nutrients are beneficial per se.

There are some drawbacks that should be evaluated, e.g. conclusions and lack of statistical analysis for the results presented in T1.

Round 2

Reviewer 1 Report

It was difficult to check the corrections as the line numbers in the manuscript text and the given answers do not match at all.

Please use in  56-58 L : „Although substrates contain nutrients that...“

Please use the author name for the first used species Latin name  (34 L).

Point 10: the equipment manufacturer (Multiskan GO, Thermo Scientific™, USA) use only when the first time mention.

Minor editing of English language required

Reviewer 2 Report

The authors have revised their manuscript followed by my comments and suggestions, and the manuscript has been much improved.

The manuscript can be accepted by this form.

English is good, but please ask English editing by native English speaker.

Author Response

We are most grateful to you for your helpful comments on the manuscript 

Reviewer 3 Report

 Most of my remarks have beend included. I have one doubt regarding Point7 and Authors' response. I am not sure (maybe it requires additional explanation) but I assume that some of data presented were not subjected under statistical analysis.

Point 7: however Table 1 lacks S.D. values and designation of statistical groups.

Response 7: We already revised following the comments and showed it in part 2.8

*I had a writing error in section 2.8. which has been revised by Point 6.

 My remarks have been included
